# Serum zinc as a biomarker to predict the efficacy of immune checkpoint inhibitors in cancers

**Jingliang Wang[1,2]☯, Weihao Wang[2]☯, Bin Liu[3], Rui Zhao[2], Jing Zhao[2], Fengxian Jiang[1,2], Wei Xu[4], Zhizhao Zhang[2], Pancen Ran[1,2], Yang Shu[1,2], Yahui Wang[1,2], Liying Pan[1,2], Lei Liu[2,5], Fang Luan[6‡], Guobin Fu[1,2,7]‡***

1 The Second Clinical Medical College, Shandong University of Traditional Chinese Medicine, Jinan, China, 2 Department of Oncology, Shandong Provincial Hospital Affiliated to Shandong First Medical University, Jinan, Shandong, China, 3 Department of Biomedical Engineering, Shandong Provincial Hospital Affiliated to Shandong First Medical University, Jinan, Shandong, China, 4 Department of Oncology, Shandong Provincial Hospital, Shandong University, Jinan, Shandong, China, 5 Department of Oncology, Jinan People's Hospital Affiliated to Shandong First Medical University, Jinan, Shandong, China, 6 Department of Clinical Laboratory, Shandong Provincial Hospital Affiliated to Shandong First Medical University, Jinan, Shandong, China, 7 The Third Affiliated Hospital of Shandong First Medical University, Jinan, China

☯ These authors contributed equally to this work.
‡ These authors also contributed equally to this work.
* fgbs@sina.com

## Abstract

### Purpose

The aim of this study was to investigate whether serum zinc levels correlate with the response to immune checkpoint inhibitors (ICIs) and whether they can be used as a useful prognostic biomarker in patients with advanced or metastatic cancer.

### Methods

We divided 98 patients with advanced or metastatic lung, esophageal, gastric, and colorectal cancer into two groups based on enrollment date: the training group (n = 68) and the validation group (n = 30). And these patients were from Shandong Provincial Hospital and had received immunotherapy. We then used the solid tumor response Evaluation Criteria (RECIST v1.1) to determine whether the patient's condition was evaluated for clinical benefit response (CBR) or non-clinical benefit (NCB). Subsequently, serum zinc levels were assessed using ICP-MS.

### Results

We have identified for the first time that elevated levels of serum zinc (>14.2μg/L) in cancer patients undergoing immunotherapy can serve as a novel biomarker for improved overall survival (20.0m vs 10.0m; p < 0.0001), as determined by continuous serum zinc data using ROC curve analysis (sensitivity: 100.00%, specificity: 41.86%, p = 0.0009) in both CBR (n = 43) and NCB patients (n = 25) within the training group.

**Data availability statement:** The data used to support the findings of this study are available from the manuscript itself

**Funding:** This work was supported by Taishan Scholar Foundation of Shandong Province [Grant/Award Number: tsqn202103179]; National Natural Science Foundation of China [Grant/Award Number: 81802284]; Science and Technology Development Plans of Shandong Province [Grant/Award Number: 2014GSF118157]; Scientific Research Foundation of Shandong Province of Outstanding Young Scientists [Grant/Award Number: BS2013YY058]; 2021 Shandong Medical Association Clinical Research Fund [Grant/Award Number: YXH2022ZX02176]; Natural Science Foundation of Shandong Province [Grant/Award Number: ZR2022MH088]; Key Development Program of Shandong Province (Grant/Award Number: 2018GSF118116), and National Natural Science Foundation of China (Grant/Award Number: 81101484).

**Competing interests:** The authors have declared that no competing interests exist.

Bioinformatics analysis has revealed that serum zinc may modulate cellular DNA replication through the MAPK and NF-kB pathways, with proteomic analysis confirming enrichment of these pathways based on KEGG and GO analyses. Consequently, a nomogram incorporating multiple clinical and independent factors has been developed to provide enhanced predictive capability.

## Conclusions

Serum zinc levels are positively associated with the effectiveness of ICIs in patients with advanced or metastatic cancer, potentially through their modulation of NF-κB and MAPK pathways. These findings highlight serum zinc as a valuable biomarker for predicting responses to ICI treatment.

---

## Introduction

Malignant neoplasms are a leading global health concern, ranking as the second highest cause of mortality [1,2]. This underscores the urgent need for innovative cancer therapeutics. ICIs, which target the interaction between programmed cell death protein 1 (PD-1) and its ligand (PD-L1), have emerged as a transformative approach in cancer treatment [3–5]. By neutralizing inhibitory signals in the T-cell environment, as well as modulating immune cells, ICIs enhances the immune system's ability to generate an anti-tumor response [6,7]. Monoclonal antibodies against PD-1 and PD-L1 are now widely used to treat various cancers, including melanoma, non-small cell lung cancer (NSCLC), and head and neck squamous cell carcinoma [8]. However, a significant proportion of patients fail to benefit from immunotherapy, with non-response rates of 52% for PD-L1-positive and 85% for PD-L1-negative individuals [9,10]. The objective response rate (ORR) to ICIs typically ranges from 20% to 40% [11], highlighting the need for predictive biomarkers to identify patients most likely to respond and to minimize adverse effects [12].

PD-L1 expression is the most widely studied biomarker, but its predictive value remains inconsistent. For example, the Keynote-024 and Keynote-189 trials demonstrated improved progression-free and overall survival in NSCLC patients with a PD-L1 tumor proportion score (TPS) above 50% treated with Pembrolizumab [13,14]. In contrast, the Checkmate-032 trial found that urothelial carcinoma patients with minimal PD-L1 expression (<1%) achieved a higher objective remission rate [15], while the Impower-133 trial reported no association between PD-L1 expression and efficacy in small cell lung cancer [16]. These discrepancies suggest that factors such as antibody variability and tumor heterogeneity influence outcomes, necessitating the discovery of alternative biomarkers to improve predictive accuracy.

Emerging evidence suggests that metal ions, particularly zinc, play a critical role in immune regulation and may influence the efficacy of ICIs. Zinc is essential for thymus function and T cell development, and its levels directly affect T helper cell balance and immune responses [17–19]. Given these findings, we hypothesize that zinc ions

modulate the tumor immune microenvironment and may serve as a predictive biomarker for ICI efficacy. This study aims to explore the relationship between zinc levels and the efficacy of immune checkpoint inhibition, providing a novel perspective for optimizing immunotherapy outcomes.

## Materials and methods

### Ethical considerations

Participants for this research were recruited from the Shandong Provincial Hospital, affiliated with Shandong First Medical University, between June 1, 2021, and February 28, 2022. The study concentrated on prevalent advanced or metastatic solid tumors, including lung and gastrointestinal cancers. Patients eligible for the study were those receiving a combination of chemotherapy and immunotherapy.

Informed consent was obtained from all participants, and the study adhered to the principles of the Declaration of Helsinki. It was approved by the Medical Ethics Committee of Shandong First Medical University, with the approval number SWYX: NO.2020−304. Confidentiality was strictly maintained, with no disclosure of patients' personal information. Participants in the retrospective analysis provided consent for their data to be utilized for research.

### Patient inclusion and treatment

A total of 98 patients with advanced cancer undergoing combination therapy at the Cancer Center of Shandong Provincial Hospital Affiliated to Shandong First Medical University were initially included in this study. The enrolled patients received ICIs, including anti-PD-1 antibody drugs such as Pembrolizumab (Keytruda, Merck), Nivolumab (Opdivo, BMS), Tislelizumab (BGB-A317, Beigene), Camrelizumab (SHR-1210, Hengrui), and Sintilimab (IBI308, Innovent); as well as anti-PD-L1 antibody drugs like Atezolizumab (Tecentriq, Roche) and Durvalumab (Imfinzi, AstraZeneca). See S1 Table for specific usage and dosage.

The inclusion criteria were adults aged 18 and above undergoing ICI combination therapy for advanced or metastatic solid tumors, with at least two accessible hematologic samples and a driver gene status that was negative or unknown. Patients were excluded if they were under 18, had multiple primary tumors, incomplete clinical data, a positive driver gene status, early-stage (I or II) cancers, or other pathological types.

### Serum preparation

Peripheral blood samples were collected from patients in the morning before each ICIs treatment using a vacuum sampling vessel (BD, 367820) on an empty stomach. Whole blood samples of serum and peripheral blood were centrifuged at room temperature (24°C, H1850R, Cence) at $1100 \pm 100G$ for 10 minutes. The serum was then extracted and stored in EP tubes (3121538, Virya), and frozen at −80°C for analysis.

The serum sample's EP tube was frozen in an upright position to ensure that the plasma remained at the bottom of the tube. Additionally, it is important to avoid repeated freezing and thawing of the plasma sample.

### Trace elements assay

Lithium (Li), magnesium (Mg), calcium (Ca), vanadium (V), chromium (Cr), manganese (Mn), iron (Fe), cobalt (Co), copper (Cu), zinc (Zn), arsenic (As), selenium (Se), strontium (Sr), molybdenum (Mo), cadmium (Cd), Hg, thallium (Tl), plumbum (Pb) and bismuth (Bi) were detected by limiting dilution method 7900 Inductively Coupled Plasma Mass spectrometry (ICP-MS) (Agilent, Tokyo, Japan). The ultrapure water (≥18.0MΩ.cm, 25°C) used in the experiment was prepared by a Milli-Q ultrapure water meter (Millipore Corporation, USA). Muti-trace elements dilution kit was used for measurement (LOT: WL210902 Baichen, Hang zhou, China). Setting parameters for the ICP-MS method were in S2 Table.

## Sample collection time and definition

T0, T1, and T2 denoted the three time points during the course of treatment, with T0 representing the pre-immunotherapy time point, T1 indicating the pre-second immunotherapy time point, and T2 corresponding to a CT evaluation prior to the third immunotherapy. All serum zinc levels were measured using ICP-MS (SIMS). Our goal was to gather serum samples from each patient before every immunotherapy session and track the fluctuations in SIMS levels during the entire duration of immunotherapy. C0 is defined as prior to cancer patients receiving their first dose of immunotherapy; C1-C7 are defined as the time points before cancer patients receive their second through eighth doses of immunotherapy.

Two independent groups in T1 were stratified based on entry time for the single-center study. Patients in the training group were enrolled from June 1 to December 31, 2021, while patients in the validation group were enrolled from January 1 to February 28, 2022.

## Clinical data and efficacy evaluation

The clinical data of patients were collected from medical records including gender, age, histology, stage, treatment lines, hematological parameters (AAT (α1-AT: Alpha 1-carbamate)), CRP (C-reactive protein), HGB (Hemoglobin), NLR (neutrophil to lymphocyte ratio).

Treatment efficacy was radiologically evaluated bi-monthly using RECIST v1.1 criteria. Complete remission (CR) was defined by the total vanishing of tumor lesions for a minimum of four weeks, while partial remission (PR) was identified by a reduction in tumor size of at least 30% over the same period. Stable disease (SD) was subcategorized into SD- for tumor shrinkage less than 30% and SD for tumor growth less than 20%. Progressive disease (PD) was diagnosed with a 20% or greater increase in tumor size. Patients were designated as having a clinical benefit response (CBR) if they experienced CR, PR, or SD with tumor reduction; those with SD without reduction or PD were labeled as having a non-clinical benefit response (NCB). The objective response rate (ORR) was calculated as the proportion of patients with a sustained reduction in tumor volume of 30% or more for at least four weeks, including those with CR and PR. PFS and OS were determined via follow-up calls and clinical data aggregation, with a cut-off date of February 31, 2022, for the assessment of non-progression, death, and survival.

## Nomogram analysis

Univariate and multivariate analysis was obtained by R language (v4.2.1; packages: "plyr", "survivalRoc", "pec") analysis, Nomogram was obtained by R language (v4.2.1; packages: "survival", "survminer") (http://www.r-project.org) analysis.

## Statistical analysis

SPSS 25.0 and Graph Pad Prism 9.0 were utilized for conducting all statistical analyses. The normality of grouped data was assessed, and the results of the normality test determined the selection of t-test, Paired t-test, or Mann-Whitney U test. Additionally, a ROC curve was constructed for the difference data. The relationship between serum trace elements and PFS or OS was evaluated using Kaplan-Meier regression, while the Nomograph was derived from Univariate and Multivariate Cox regression analyses. A p value < 0.05 was considered to indicate statistical significance. Integration of database and Nomograph association employed R (v4.2.1).

## Results

### 1. Patients

From a larger pool of 233 patients with advanced or metastatic tumors, 98 who met the study's inclusion criteria were enrolled. These participants, exhibiting diverse treatment durations, were stratified into groups for serum trace element analysis via ICP-MS, as illustrated in Fig 1. Demographic and baseline hematological data for this cohort, extracted from

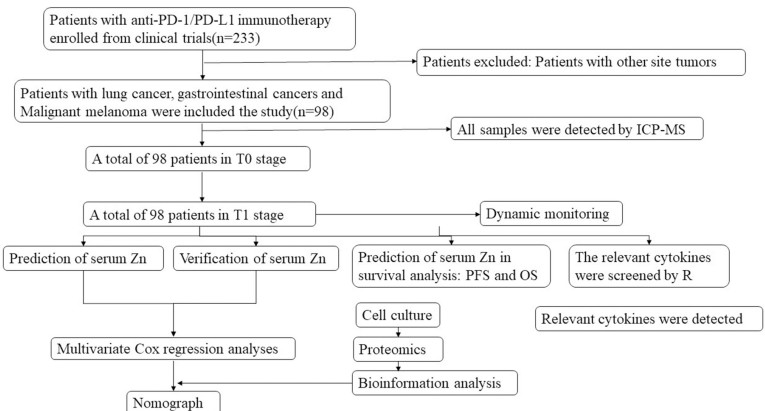

**Fig 1. Study flow chart.** Study workflow. Patients with advanced or metastatic cancer at the time of enrollment were screened according to established conditions.

medical records, are detailed in Tables 1 and 2. Upon study entry, all patients were undergoing concurrent chemotherapy and immunotherapy. The average age was over 60 years, with the CBR group averaging 63.03±9.566 years and the NCB group 59.54±10.24 years. The cohort included 24.5% females and 75.5% males, predominantly with lung and gastro-intestinal cancers. The majority, 65.3%, were on their initial round of immunotherapy, and for 27.6%, PD-L1 expression levels were precisely documented. Notably, variations in the number of treatment lines and PD-L1 expression were the only variables that reached statistical significance among those analyzed.

In addition, a literature review revealed acute phase proteins AAT and CRP as potential indicators of immune status, and NLR as a potential independent predictor of immunotherapy prognosis. Based on this, we selected these three hematologic markers, which are associated with immunotherapy efficacy and zinc metabolism, for our analysis. Baseline assessments indicated no significant differences in AAT levels between the CBR group (179.6±43.87) and NCB group (197.6±48.16; p=0.0876), nor in CRP levels (CBR: 12.96±24.07, NCB: 17.73±27.54; p=0.4506), and NLR values (CBR: 2.512±2.138, NCB: 2.384±1.685; p=0.9195), indicating similar initial measurements for these parameters in both groups.

## 2. Serum trace element levels in different stages showed different predictive abilities

**2.1. Serum trace element level at T0 was used as the baseline to show efficacy of ICIs.** Prior to the commencement of immune checkpoint inhibitor (ICI) therapy, we evaluated the serum trace element concentrations in 98 patients, divided into clinical benefit response (CBR) and non-clinical benefit (NCB) groups, to explore the potential influence of initial trace element levels on immunotherapy outcomes. Our analysis did not identify any significant disparities in serum trace element profiles between the CBR group (n=62) and NCB group (n=36). This lack of distinction at baseline leads us to suggest that variations in serum trace element levels could be a consequence of immunotherapy rather than a predisposing factor (S3 Table).

**2.2. The Zn predicts response to ICIs at 3 weeks (T1) on treatment.** At the T1 stage, serum trace element levels were analyzed for 98 patients, segregated into a Training group (43 CBR and 25 NCB patients) and a validation group (19 CBR and 11 NCB patients). A marked difference in zinc levels was identified between the NCB and CBR groups in the Training set (p=0.0007). The ROC curve analysis revealed that zinc was a potent predictor of immunotherapy efficacy, exhibiting a sensitivity of 100.00% and specificity of 41.86% (p=0.0009), with a determined threshold of 14.2 µg/ml for cancer patients (Fig 2A). In the validation cohort, the true positive rate was 73.00%, and the true negative rate was 37.00%. These findings underscore the potential utility of zinc as a predictive biomarker in immunotherapy.

**Table 1. Comparison of baseline parameters between CBR and NCB.**

|  |  | CBR (n=62) | NCB(n=36) | P value |
|---|---|---|---|---|
| Age (Year) | Mean±SD | 63.03±9.566 | 59.54±10.24 | 0.1451 |
| Gender | Female | 13 | 11 | 0.3338 |
|  | Male | 49 | 25 |  |
| Position | Lung | 32 | 19 | >0.9999 |
|  | Digestive tract | 30 | 17 |  |
| Metastasis | M0 | 28 | 20 | >0.9999 |
|  | M1 | 34 | 18 |  |
| Treatment lines | First-line treatment | 53 | 11 | <0.0001 |
|  | Posterior-line therapy | 9 | 25 |  |
| PD-L1 expression | Positive | 20 | 7 | 0.2413 |
|  | Negative or Unavailable | 42 | 29 |  |

**Table 2. Comparison of hematological baseline parameters between CBR and NCB.**

|  | CBR(n=62) | NBC(n=36) | P value |
|---|---|---|---|
| AAT(mg/dl) | 179.6±43.87 | 197.6±48.16 | 0.0876 |
| CPR(mg/L) | 12.96±24.07 | 17.73±27.54 | 0.4506 |
| NLR | 2.512±2.138 | 2.384±1.685 | 0.9195 |

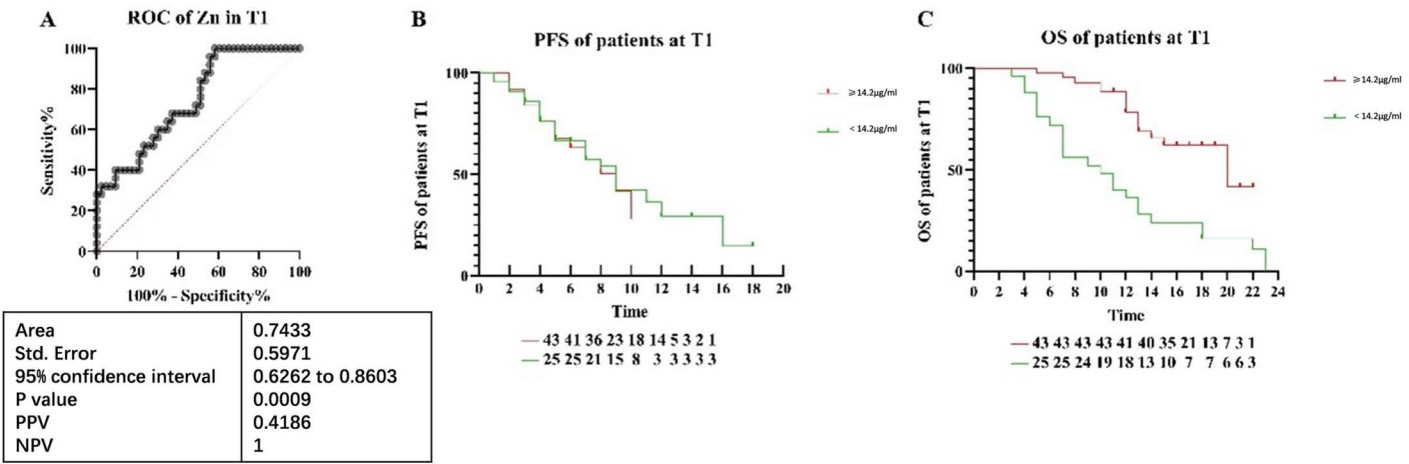

**Fig 2. ROC curves in different groups of patients, and PFS and OS of patients. (A)** ROC curve of Zn predicting immunotherapy response in all tumor types. **(B)** The PFS was depicted grouping by the CUT OFF value in cancers (p>0.05). **(C)** The OS was depicted grouping by the CUT OFF value in all tumor types (p<0.0001).

Our subsequent observations indicated no significant difference in progression-free survival (PFS) between patients with zinc levels above the established threshold and those below it (median PFS of 9.0 months for both; p=0.7190, Fig 2B). In contrast, a pronounced improvement in overall survival (OS) was noted in patients with higher zinc levels, with a median OS of 20.0 months compared to 10.0 months for those with lower levels (p<0.0001, Fig 2C). These results suggest that while zinc levels do not predict PFS, they may be a significant predictor of OS in cancer patients undergoing immunotherapy.

## 3. The relationship between zinc and immune-related genes

We explored the influence of serum zinc on immunotherapy outcomes through an in vitro experiment using the A549 lung adenocarcinoma cell line. The Zn2+-induced IC50 was determined to be 1.2μM through the CCK8 assay (S1 Fig). Despite potential access issues to the GEO database website, we analyzed gene expression data from ICI-treated patients. DEGs were evaluated through KEGG and GO enrichment analyses, and a 4D label-free proteomics analysis was conducted on A549 cells to further investigate the impact of Zn2+. For the quantification of significant differences, proteins with a fold change over 2 and a p-value under 0.05 were identified as differentially expressed. Our analyses revealed that Zn2+ is significantly associated with the MAPK and NF-kB pathways, as well as DNA replication, corroborating bioinformatics findings from the GEO database (Fig 3A,3B). The integration and enrichment of omics data from the GEO immunotherapy prognostic database led to the discovery of 47 differential genes (Fig 3C). This evidence supports the hypothesis that increased Zn2+ levels correlate with improved tumor suppression in cancer patients.

## 4. Multi-factor prediction model

Trace elements have been recognized for their significant predictive power concerning the success of immunotherapy. To capitalize on this, we integrated trace elements with other relevant metrics to develop a comprehensive model. After initial testing of the indicators, we formulated a Nomogram that estimates progression-free survival (PFS) in immunotherapy,

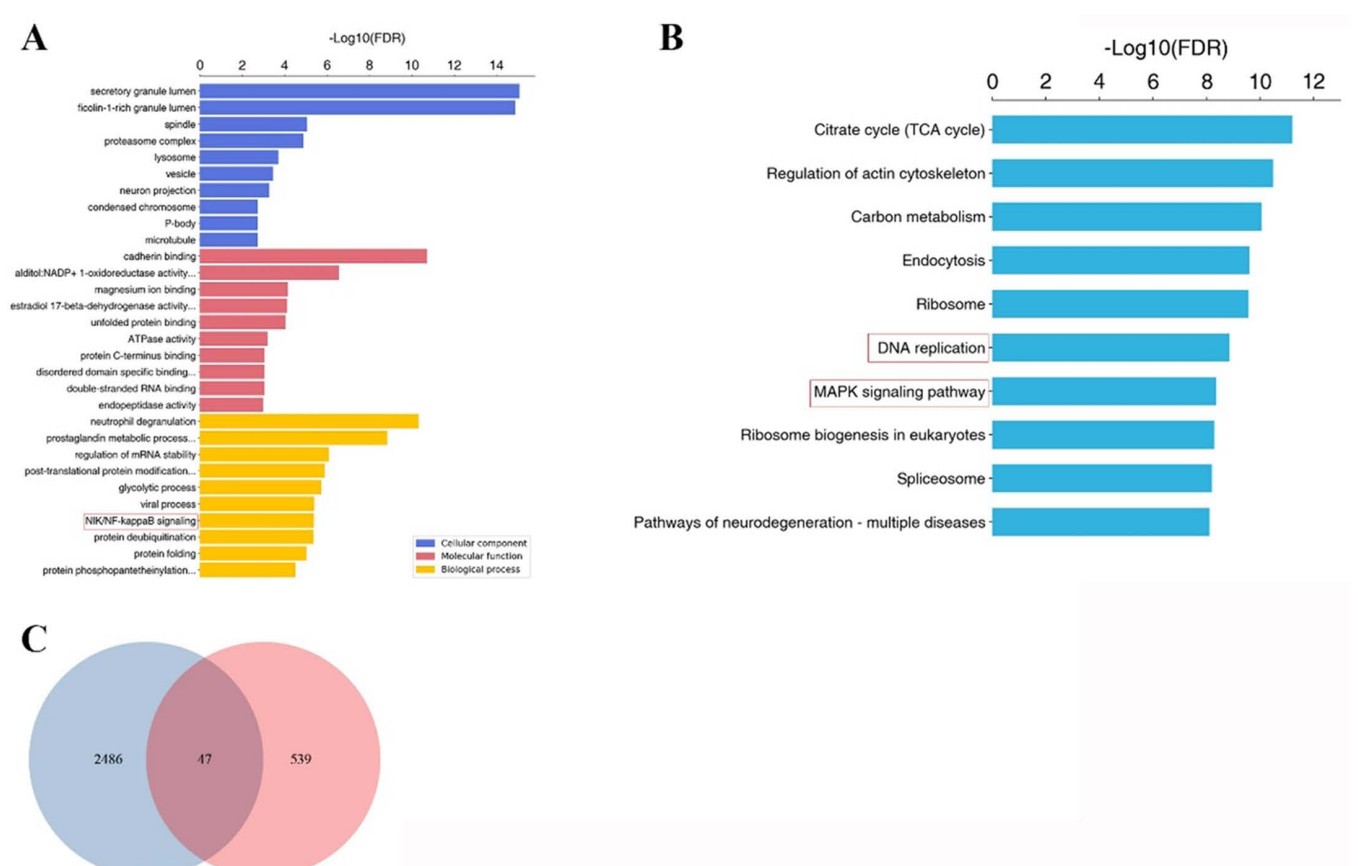

**Fig 3. Biological information suggests pathway changes. (A)** GO analysis of proteomics. **(B)** KEGG analysis of proteomics. **(C)** Venn of Immunity and proteomics.

with C-reactive protein (CRP) as a risk factor and zinc (Zn) as a protective factor (Fig 4A). We also developed a parallel Nomogram to predict overall survival (OS), using identical factors (Fig 4D). The results showed that patients with low CRP levels or high serum zinc levels had lower total scores. These patients have a longer harvest of PFS and OS after immunotherapy. To assess the predictive accuracy of these models, calibration curves were plotted (Figs 4B, 4C, 4E, 4F). These curves can be used to assess the degree of model calibration by judging whether the model output probability is consistent with the actual observed probability to ensure the reliability of our nomogram in predicting immunotherapy outcomes.

## Discussion

There is an urgent need for a sensitive predictive biomarker to assess treatment outcome in the Asian patient population with advanced cancer currently treated with ICIs. Our study demonstrates that serum zinc, as an easily detectable, non-invasive biomarker, can be used as a reliable predictor of response to immunotherapy in various cancers. Therefore, we recommend the assessment of serum zinc levels in patients receiving immunotherapy. Additionally, we conducted univariate and multivariate Cox regression analyses on all parameters, followed by the creation of a nomogram and calibration curves to evaluate the predictive accuracy of our model.

Our study benefits significantly from the deployment of advanced detection methods, particularly the use of inductively coupled plasma-mass spectrometry (ICP-MS) for the analysis of serum zinc. This technique provides a comprehensive assessment of total serum zinc, including both protein-bound and free zinc, offering insights into the body's zinc reserves. ICP-MS surpasses traditional methods in precision and accuracy for zinc detection. Importantly, serum zinc level

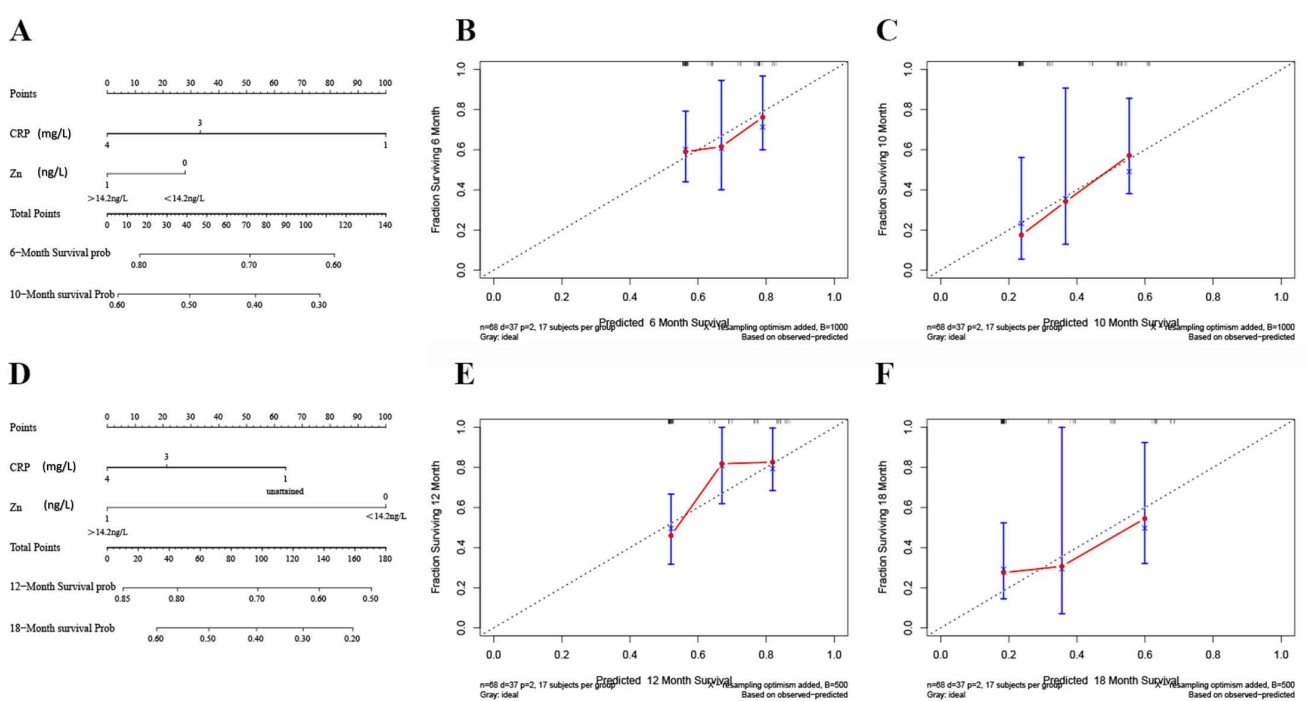

**Fig 4. Nomograph and calibration curve. (A)** The nomogram to predict the efficacy of immunotherapy was created to PFS. **(B)** The calibration curve for the prediction of the efficacy of immunotherapy in 6 months. **(C)** The calibration curve for the prediction of the efficacy of immunotherapy in 10 months. **(D)** The nomogram to predict the efficacy of immunotherapy was created to OS. **(E)** The calibration curve for the prediction of the efficacy of immunotherapy in 12 months. **(F)** The calibration curve for the prediction of the efficacy of immunotherapy in 18 months.

assessment is widely accessible in clinical settings, is less invasive, cost-effective, and has the potential to predict the effectiveness of ICIs at an early stage, thereby enhancing treatment strategies.

Research by Dr. Lossow has demonstrated that cancer patients typically exhibit lower serum zinc levels compared to healthy individuals, while zinc levels within tumors remain unchanged. This observation suggests that elevated pre-treatment serum zinc levels may correlate with improved prognoses following immunotherapy [20]. The underlying mechanisms for this correlation may involve zinc's critical role in immune regulation. For instance, zinc is essential for T cell development, differentiation, and activation [21]. Moderate zinc deficiency (3–5 mg/day) has been linked to compromised Th1 and Th2 cell functions, leading to a reduced cell-mediated immune response [22]. Furthermore, zinc deficiency can exacerbate oxidative stress, resulting in increased oxidative DNA damage, altered gene expression, and potential harm to tissues or organs [23,24]. These findings highlight the importance of zinc homeostasis in maintaining immune function and optimizing cancer treatment outcomes.

Zinc's immunomodulatory properties extend to its ability to influence specific immune pathways. For example, zinc sulfate at a concentration of 50 μM has been shown to enhance Th1 cytokine production while mitigating Th2-driven chronic anaphylaxis, thereby inhibiting the expansion of allergen-activated peripheral blood mononuclear cells [25]. In animal models, dietary zinc supplementation (95 mg/kg) downregulated interferon-gamma mRNA expression in the airways of asthmatic rats, underscoring zinc's role in modulating respiratory immune responses [26]. These studies collectively suggest that zinc supplementation could serve as a therapeutic strategy for allergic diseases and asthma by restoring immune balance.

In a mouse drug-induced lung tumor model, it has been observed that activation of the NF-κB pathway in the airway epithelium induces chronic inflammation and recruits regulatory T cells to promote lung tumor formation [27]. Persistent NF-κB activation also contributes to chemoresistance in lung tumor cells to platinum-based drugs and other chemotherapeutics, further complicating treatment efficacy [28,29]. Interestingly, intraperitoneal zinc administration (10 mg/kg) enhances the function of zinc finger protein A20, which suppresses NF-κB signaling through receptor-interacting protein 1 (RIP1) degradation. This mechanism attenuates tumor aggressiveness and metastasis, highlighting zinc's potential as an adjunct therapy in cancer treatment [30].

While our findings provide valuable insights, several limitations must be acknowledged. The single-center design of our study may have introduced selection bias, limiting the generalizability of our results. Moreover, on the one hand, we cannot control dietary intake. On the other hand, because of laboratory limitations, we were unable to measure antioxidant protein levels, and these changes may affect serum micronutrient concentrations [31,32]. The temporal confounding of treatment-induced oxidative stress was minimized by measuring baseline serum zinc levels. In the future, these limitations will be addressed by incorporating a multicenter design, greater follow-up, and improved testing. Furthermore, we plan to investigate the impact of adverse reactions, particularly diarrhea, on zinc levels, given the prevalence of zinc malabsorption in patients with intestinal mucosal damage and chronic diarrhea [33].

Our study provides preliminary evidence that there is a correlation between serum zinc levels and the efficacy of immunotherapy. This correlation is reflected in treatment response and overall survival in patients with advanced cancer. These findings not only underscore the importance of zinc in immune regulation and cancer therapy but also pave the way for developing predictive biomarkers and synergistic treatment strategies to enhance immunotherapy outcomes.

## Supporting information

**S1 Table.  Usage and dosage of immune checkpoint inhibitors.**
(DOCX)

**S2 Table.  Setting parameters for the ICP-MS method.**
(DOCX)

**S3 Table. The serum trace element levels of cancer patients enrolled before the first treatment(T0).** *Detection of baseline levels of 10 selected trace elements.*
(DOCX)

**S1 Fig. A549 Cell survival was measured by CCK8.**
(TIF)

## Author contributions

**Data curation:** Bin Liu, Jing Zhao, Yang Shu, Rui Zhao, Liying Pan.

**Formal analysis:** Fengxian Jiang, Wei Xu, Zhizhao Zhang, Pancen Ran, Yahui Wang, Lei Liu.

**Writing – original draft:** Jingliang Wang, Weihao Wang.

**Writing – review & editing:** Fang Luan, Guobin Fu.

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
