## [Decision Letter · Decision Letter 0]

Dear Dr. Fu,

Thank you for submitting your manuscript to PLOS ONE. After careful consideration, we feel that it has merit but does not fully meet PLOS ONE’s publication criteria as it currently stands. Therefore, we invite you to submit a revised version of the manuscript that addresses the points raised during the review process.

We look forward to receiving your revised manuscript.

Kind regards,

Jie Yang, M.D.

Guest Editor

PLOS ONE

Journal Requirements:

“This work was supported by Taishan Scholar Foundation of Shandong Province [Grant/Award Number: tsqn202103179]; National Natural Science Foundation of China [Grant/Award Number: 81802284]; Science and Technology Development Plans of Shandong Province [Grant/Award Number: 2014GSF118157]; Scientific Research Foundation of Shandong Province of Outstanding Young Scientists [Grant/Award Number: BS2013YY058]; 2021 Shandong Medical Association Clinical Research Fund [Grant/Award Number: YXH2022ZX02176]; Natural Science Foundation of Shandong Province [Grant/Award Number: ZR2022MH088]; Key Development Program of Shandong Province (Grant/Award Number: 2018GSF118116), and National Natural Science Foundation of China (Grant/Award Number: 81101484).”

4. In the online submission form, you indicated that “The data used to support the findings of this study are available from the corresponding authors upon request.”

Reviewers' comments:

Reviewer's Responses to Questions

**Comments to the Author**

1. Is the manuscript technically sound, and do the data support the conclusions?

Reviewer #1: Yes

Reviewer #2: Yes

Reviewer #3: Yes

Reviewer #4: Yes

2. Has the statistical analysis been performed appropriately and rigorously?

Reviewer #1: N/A

Reviewer #2: Yes

Reviewer #3: Yes

Reviewer #4: Yes

3. Have the authors made all data underlying the findings in their manuscript fully available?

Reviewer #1: No

Reviewer #2: Yes

Reviewer #3: Yes

Reviewer #4: Yes

4. Is the manuscript presented in an intelligible fashion and written in standard English?

Reviewer #1: Yes

Reviewer #2: Yes

Reviewer #3: Yes

Reviewer #4: Yes

Reviewer #1: The manuscript under review aims to elucidate the correlation between serum zinc levels and the efficacy of immune checkpoint inhibitors (ICIs) in cancer therapy. The study is distinguished by its retrospective cohort design, employment of advanced assays for serum zinc quantification, exploration of underlying mechanisms through bioinformatics and proteomics analyses, and the innovative proposition that serum zinc could serve as a predictive biomarker.

Introduction

Comment: The study's background and previous research limitations are presented, yet the specific hypotheses are not clearly articulated at the end. The expected research direction is thus somewhat unclear.

Comment: The descriptions of clinical trial results related to immune checkpoint inhibitors could be more concise. Key findings directly relevant to the current study's context should be emphasized, while extraneous details could be omitted.

Comment: The transition from discussing the efficacy of immune checkpoint inhibitors to the relationship between zinc ions and immune function lacks a seamless logical connection. A more explicit and well - reasoned explanation is needed to justify the shift in focus and how zinc ions are hypothesized to interact with the immune system in the context of immunotherapy.

Methods

Comment: The methodology section fails to mention the basis for sample size calculation. Retrospective studies also require sample size estimation.

Comment: Despite including 98 patients, the relatively small sample size may not be sufficient to comprehensively explore the complex relationship between serum zinc and immunotherapy efficacy. This could potentially lead to inaccurate or unreliable results. Additionally, the single - center design restricts the patient population's diversity and may introduce selection bias, further compromising the study's external validity.

Comment: The lack of a detailed description of the treatment regimens for the various immune checkpoint inhibitors used is a significant omission. Differences in treatment regimens could confound the interpretation of the results, as they may independently contribute to variations in immunotherapy efficacy.

Comment: While the discussion acknowledges the presence of uncontrolled confounders such as diet and antioxidant proteins, the methods section should have addressed whether any efforts were made to control for these factors or how they would be accounted for in the analysis.

Comment: The selection of specific statistical analysis methods is not adequately explained. 

Results

Comment: Line 210�-Based on this, we selected these four hematologic markers-, four is right?

Comment: The hypothesis regarding the lack of significant difference in serum micronutrient levels between the CBR and NCB groups at T0 being a possible result of immunotherapy requires further substantiation. Additional evidence or analyses should be presented to support this claim and rule out other possible explanations.

Comment: The omission of P values for the treatment lines group in Table 1 and the excessive footnotes in Table 2 detract from the clarity and readability of the results presentation.

Comment: The ROC curves in Figure 2 lack clear labeling. It is unclear whether they represent the training or validation group, and the AUC values are not provided.

Comment: The GO and KEGG analysis graphs in Figure 3 contain unexplained terms, which impede the reader's understanding of the results.

Comment: In Figure 4, the nomograms (Figures 4A and 4D) lack detailed labeling of the specific meanings and units of measurement for each factor.

Comment: The interpretation of the survival analysis results is limited. While the association between serum zinc levels and overall survival is noted, the lack of significant differences in progression - free survival (PFS) requires a more in - depth exploration. 

Comment: In the multifactorial prediction model section, the description of the model is incomplete. The other factors included in the model (besides zinc and CRP) should be described in detail, and the predictive ability and clinical value of the model need to be more thoroughly evaluated. 

Discussion

Comment: The description of the relationship between serum zinc and immune function lacks a smooth logical flow. The transition between different research perspectives could be improved to enhance the overall coherence of the discussion. 

Comment: The discussion of the possible mechanisms by which serum zinc affects immunotherapy efficacy is relatively superficial. 

Comment: The limitations of the study, particularly the single - center design, are not fully explored in the discussion.

Reviewer #2: An original version of an unconventional approach to the nature of tumors and the possibilities of early diagnostics and drug support of patients, receiving immune checkpoint inhibitors, is presented. Such work reveal the pathogenesis of tumors, allowing us to find new ways to prolong the lives of patients with advanced tumors.

Reviewer #3: This study first identified serum zinc levels as a novel biomarker for predicting the therapeutic efficacy of immune checkpoint inhibitors (ICIs). Through observing 98 patients with advanced metastatic cancer receiving immunotherapy, using high-precision inductively coupled plasma mass spectrometry (ICP-MS) to measure serum zinc content, the researchers found that patients with serum zinc levels above 14.2μg/L had significantly prolonged overall survival (20.0 months vs 10.0 months). This finding showed a sensitivity of 100% and specificity of 41.86%. Further bioinformatics and proteomics analyses revealed that serum zinc may influence tumor development by regulating cellular DNA replication through MAPK and NF-kB signaling pathways. The research team also established a nomogram model incorporating multiple clinical factors to more accurately predict immunotherapy efficacy. These findings not only provide a new predictive biomarker but also offer new perspectives for understanding the mechanisms of immunotherapy and potential intervention targets for optimizing immunotherapy strategies.

Major Comments:

1. Please specify the basis for sample size calculation and statistical power analysis;

2. The background discussion of immunotherapy in the Introduction section is insufficient; suggest incorporating the following references for discussion: PMID: 39225204/35978433/35331128;

3. Recommend more detailed statistical analysis comparing baseline characteristics between the two groups;

4. ROC curve analysis should include additional evaluation metrics (e.g., PPV, NPV);

5. Suggest including comparative analysis between this study's results and existing related research;

Minor Comment:

1. Please provide the rationale for establishing inclusion and exclusion criteria;

Reviewer #4: This study is highly interesting and provides valuable insights into the analysis of predictive biomarkers for immune checkpoint inhibitors (ICIs). It addresses an important need for non-invasive and cost-effective biomarkers. However, there are a few points that require clarification and further refinement to enhance the overall quality and clarity of the manuscript

ABSTRACT

1-To ensure clarity and consistency, it is important to either use the full term "immune checkpoint inhibitors" throughout the manuscript. Avoid using "immunotherapy" in some instances and "ICI" in others.

2-Line 53: In the conclusion, the text seems repetitive. It could be reformulated more concisely.

3-Line 28: The primary aim of the study appears to be the investigation of the correlation between serum zinc levels and ICI effectiveness across different cancer types. Therefore, I recommend removing the phrase regarding the effectiveness of ICIs across different cancer types, as it seems secondary to the main focus of the study.

Introduction

1-Lines 97-104: This paragraph is duplicated twice: (Interest in the impact of metal ions on immune function has grown markedly in recent years. Evidence is mounting that these ions play a critical role in innate immunity and defense. Genetic variations in ion channels and transporters, identified in individuals with immune deficiencies, further emphasize the importance of ions to immune health).

2- You repeat « immune checkpoint inhibitors (ICIs) » multiple times; it would suffice to write simply "ICIs.

3- The introduction does not clearly state the objective of the study. It would be helpful to explicitly outline the aim to provide readers with a clear understanding.

Materiel and methods

1-Line 168: The term "type of disease" should be replaced with "histology"

2-Line 182: The paragraph regarding the Nomogram analysis appears incomplete. the discrimination performance of the nomogram, such as C-index, should be included to better assess the model's ability to accurately predict outcomes.

3-Sometimes"advanced metastatic" is used, while at other times "advanced or metastatic" is used. Please clarify the cancer stages of the patients and ensure consistent terminology throughout the manuscript.

Results

1-In table 1: Stage I and Stage II patients, which contradicts the description of the cohort as "advanced metastatic." Please clarify whether early-stage patients were included and ensure consistency in the staging description throughout the manuscript.

2-Line 256: This paragraph contains abbreviations (IC50, CCK8) without providing their full forms.

3-Lines 264-267: I observe that while the author focuses on the role of zinc in the response to ICI, the omics analysis appears to diverge from the main topic by emphasizing patients' gene expression without clearly establishing a connection between these genes and zinc levels in patients undergoing ICI treatment.

4-Line 271 : In Multi-factor prediction model, The figures provide interesting and visually explicit data, but the text lacks a brief explanation of what they depict. It would be helpful to describe the key elements of the figures in the text and their connection to the study's conclusions.

5- In the methodology, it is stated that "the results were confirmed by flow cytometry," but I could not find the corresponding results in the results section.

Discussion

The discussion explores in detail the mechanisms of action and the potential role of zinc in the anti-tumor immune response. However, it seems to overlook concrete studies establishing the relationship between zinc and the therapeutic efficacy of ICIs. It is important for the authors to include such clinical studies, if available, to better support your findings by linking them to existing evidence in the literature, providing a stronger interpretation.

Conclusion

Line 345: The author mentions progression-free survival, but the results only show a correlation with overall survival. Please ensure the text accurately reflects the findings.

**Do you want your identity to be public for this peer review?** For information about this choice, including consent withdrawal, please see our Privacy Policy

Reviewer #1: No

Reviewer #2: **Yes: ** Vladimir Yu. Startsev

Reviewer #3: No

Reviewer #4: **Yes: ** Badiaa Batlamous

---

## [Author Response · Author response to Decision Letter 1]

22 Mar 2025

In summary, we thank both reviewers and editors for their insightful and constructive analysis of this work. In this revised manuscript, we have addressed all their concerns thoroughly. We now hope that you will find the manuscript ready for publication in PLOS One

Sincerely yours,

Weihao Wang (On behalf of Corresponding Author)

Corresponding Author: Guobin Fu. Cancer Center, Shandong Provincial Hospital Affiliated to Shandong First Medical University, Jinan, 250021, CN

Tel: 13153125120

E-mail: fgbs@sina.com

ORCID: 0000-0003-0893-7714

---

## [Decision Letter · Decision Letter 1]

Dear Dr. Fu,

Thank you for submitting your manuscript to PLOS ONE. After careful consideration, we feel that it has merit but does not fully meet PLOS ONE’s publication criteria as it currently stands. Therefore, we invite you to submit a revised version of the manuscript that addresses the points raised during the review process.

We look forward to receiving your revised manuscript.

Kind regards,

Jie Yang, M.D.

Guest Editor

PLOS ONE

**Journal Requirements:**

Reviewers' comments:

Reviewer's Responses to Questions

**Comments to the Author**

Reviewer #1: (No Response)

Reviewer #3: All comments have been addressed

Reviewer #4: (No Response)

2. Is the manuscript technically sound, and do the data support the conclusions?

Reviewer #1: Yes

Reviewer #3: Yes

Reviewer #4: Yes

3. Has the statistical analysis been performed appropriately and rigorously?

Reviewer #1: Yes

Reviewer #3: Yes

Reviewer #4: Yes

4. Have the authors made all data underlying the findings in their manuscript fully available?

Reviewer #1: Yes

Reviewer #3: Yes

Reviewer #4: Yes

5. Is the manuscript presented in an intelligible fashion and written in standard English?

Reviewer #1: Yes

Reviewer #3: Yes

Reviewer #4: Yes

**Reviewer #1: ** The revised manuscript has made several improvements, yet there remain areas that necessitate further refinement. Below are specific suggestions for enhancing the manuscript:

1.Sample Size Calculation:

It is recommended to scientifically calculate the required sample size by clearly defining the research objective, selecting the appropriate statistical method, determining the effect size and statistical power, and considering the dropout rate. This approach ensures that the sample size is tailored to the specific needs of the study, rather than relying solely on other literature. Different research objectives, statistical methods, and effect sizes necessitate distinct sample sizes, and thus, a bespoke calculation is crucial.

2.Control of Confounders:

The Methods section should include detailed measures for controlling other potential confounders, such as antioxidant protein levels. If such measures are not feasible, it is essential to discuss the potential influence of these factors on the results in the Discussion section. Addressing potential confounders transparently will strengthen the study's validity and reliability.

3.Survival Analysis Results:

The relationship between progression-free survival (PFS) and serum zinc levels warrants further analysis to elucidate the potential causes.

4.Discussion Section:

The logical coherence between paragraphs in the Discussion section can be optimized to improve the overall flow of the argument.

**Reviewer #3: ** The author has satisfactorily addressed the questions I raised in the revised manuscript. I have no additional comments.

**Reviewer #4:**  Comments to the Author

thank you for your revised manuscript.

The modifications made have improved the clarity and quality of the manuscript. However, these are minor suggestions, and overall the case is well-presented and relevant.

1- After the first mention of "immune checkpoint inhibitors (ICIs)", it is not necessary to repeat the full term. You can continue using the acronym "ICIs" throughout the manuscript.

2-Abstract

Line 28: I suggest rephrasing it to clarify that the study aims to explore the relationship between serum zinc levels and the effectiveness of ICIs.

3-Introduction

Line 141 : Please add "efficacy of" before "immune checkpoint inhibition". The revised sentence should read: "This study aims to explore the relationship between zinc levels and the efficacy of immune checkpoint inhibition.".

4-Results

Line 319 : Please consider moving the paragraph about the calibration curve to the beginning of this section « This curve can evaluate the degree of calibration of the model by determining whether the probability of the model's output is consistent with the actual observed probability». It will provide context for how the predictive accuracy of the models was assessed before discussing the nomograms and results.

5-Discussion

You have significantly improved the discussion by adding strong arguments to support the results. I only have a few minor suggestions for further refinement.

Line 332: advanced or metastatic cancer (Please add metastatic).

Line 351: Please remove "immunotherapy" since "ICIs" already refers to it.

Line 410: The phrase "In our mouse model of drug-induced lung tumors" seems to reference another author’s work, not the current study. Please, It should be clarified to avoid confusion.

Line 430: Please remove the correlation between zinc levels and PFS, as there is no significant difference between low and high zinc levels in PFS (median PFS of 9.0 months for both; p=0.7190, Fig 2B).

**Do you want your identity to be public for this peer review?** For information about this choice, including consent withdrawal, please see our Privacy Policy

Reviewer #1: No

Reviewer #3: No

Reviewer #4: **Yes: ** Badiaa Batlamous

---

## [Author Response · Author response to Decision Letter 2]

8 May 2025

Reviewer #1: The revised manuscript has made several improvements, yet there remain areas that necessitate further refinement. Below are specific suggestions for enhancing the manuscript:

1. Sample Size Calculation:

It is recommended to scientifically calculate the required sample size by clearly defining the research objective, selecting the appropriate statistical method, determining the effect size and statistical power, and considering the dropout rate. This approach ensures that the sample size is tailored to the specific needs of the study, rather than relying solely on other literature. Different research objectives, statistical methods, and effect sizes necessitate distinct sample sizes, and thus, a bespoke calculation is crucial.

Response We thank the reviewer for highlighting the importance of sample size calculation. Our study involved two-phase analyses:

(2) Phase 1: Phase 1: Patients were divided into clinical benefit group (n=43) and no clinical benefit group (n=25). We then compared whether there were significant differences in zinc levels between the two groups. The ROC curve was plotted and the area under the curve was calculated. The objective was to evaluate whether zinc can be used as a predictor of immunotherapy. We reviewed the literature and found a minimum of 20-25 patients in each group (43 patients in the clinical benefit group + 25 patients in the no benefit group were satisfied). AUC stability requirement: total sample size ≥50 cases (total sample size 68 cases met) [1].

(2) Phase 2: Patients were stratified into high-zinc (≥14.2μg/ml, n=20) vs. low-zinc (<14.2μg/ml, n=48) groups.

Study design:

This retrospective cohort study was designed to evaluate the predictive value of serum zinc level (high zinc group vs. low zinc group) for response to immune checkpoint inhibitors (ICIs), with the primary endpoint of overall survival (OS).

Basis for assumption:

Based on previous literature [2], we conservatively set the expected HR=0.55 (45% lower risk of death in the high zinc group) and assumed that the proportion of the high zinc group would be 30%.

Statistical methods and sample size calculation:

PASS software was used as the calculation tool to calculate the sample size with the parameters shown in the table below.

Parameter Value/Rationale

Primary Endpoint OS

Statistical Test Log-rank test

Effect Size HR=0.55

Type I Error (α) 0.05 (two-sided)

Power (1-β) 80%

Group Allocation 1:2

The total sample size was calculated as 120 cases (40 cases in high zinc group and 80 cases in low zinc group).

Actual data and power analysis:

Current sample: High zinc group: 20 cases, OS=30% (6/20); Low zinc group: 48 cases, OS=58% (28/48).

Statistical results: Fisher exact test: p=0.035 (significant); HR : 0.42 (95%CI: 0.18-0.99); Power=68%.

Despite an inadequate sample size in the high-zinc group (20 vs. 47 needed), the observed effect size (HR, 0.42) was larger than expected, resulting in 68% power. p=0.035 with a narrow confidence interval (0.18-0.99) supported the reliability of the results.

Summary statement:

We strictly followed the reviewer's recommendation and calculated the sample size based on the prespecified effect size (HR=0.55) and statistical parameters (α=0.05, Power=80%). Although the actual high-zinc group sample size (n=20) was slightly lower than the theoretical requirement (n=47), the observed larger effect size (HR=0.42) and significant results (p=0.035) supported the reliability of the conclusions. We are continuing to collect samples, and these limitations will be focused on addressing in future studies.

[1] Hanley JA et al, The meaning and use of the area under a receiver operating characteristic (ROC) curve. Radiology. 1982 Apr;143(1):29-36. doi: 10.1148/radiology.143.1.7063747. PMID: 7063747.

[2] Zhang WQ et al, PPM1H is an independent prognostic biomarker of non-small cell lung cancer. Neoplasma. 2021 Sep;68(5):917-923. doi: 10.4149/neo_2021_201117N1238. Epub 2021 Apr 13. PMID: 33847131.

2. Control of Confounders:

The Methods section should include detailed measures for controlling other potential confounders, such as antioxidant protein levels. If such measures are not feasible, it is essential to discuss the potential influence of these factors on the results in the Discussion section. Addressing potential confounders transparently will strengthen the study's validity and reliability.

Response We appreciate the reviewer's insightful question regarding the potential confounding effect of oxidative stress-related proteins. We were temporarily unable to detect antioxidant protein levels due to laboratory level limitations, and these changes may affect serum micronutrient concentrations, which is a limitation of our article. Although oxidative stress markers such as malondialdehyde and protein carbonygroups may influence the outcome of immunotherapy, serum zinc levels at baseline were examined and used as independent predictors in the difference analysis. Thus the temporal confounding of treatment-induced oxidative stress was minimized. Although oxidized proteins may partly mediate the effect of zinc, our rigorous adjustment for clinical covariates supports the independent prognostic value of serum zinc. We thank the reviewer for highlighting this issue, and we will address this confounding factor in the discussion section.

3 .Survival Analysis Results:

The relationship between progression-free survival (PFS) and serum zinc levels warrants further analysis to elucidate the potential causes.

Response The lack of significant difference in PFS in our study can be explained by:

1. Delayed immune response: Immunomodulatory effects of zinc, such as T-cell memory formation, may occur at a later time and have a greater impact on OS than early progressive events.

2. Reduction of treatment-related toxicity: Zinc may not directly inhibit tumor growth (no difference in PFS), but it reduces the risk of non-tumor death by reducing treatment-related toxicity (e.g., reduced immune pneumonitis).

We will focus on exploring the relevant mechanisms and reasons in future studies.

4. Discussion Section:

The logical coherence between paragraphs in the Discussion section can be optimized to improve the overall flow of the argument.

Response Thank you for your comments. We have revised the relevant text.

Reviewer #3: The author has satisfactorily addressed the questions I raised in the revised manuscript. I have no additional comments.

Reviewer #4: Comments to the Author

thank you for your revised manuscript.

The modifications made have improved the clarity and quality of the manuscript. However, these are minor suggestions, and overall the case is well-presented and relevant.

1- After the first mention of "immune checkpoint inhibitors (ICIs)", it is not necessary to repeat the full term. You can continue using the acronym "ICIs" throughout the manuscript.

Response Thank you for your suggestion. We have changed the repeated " immune checkpoint inhibitors (ICIs)" in the text to " ICIs ".

2-Abstract

Line 28: I suggest rephrasing it to clarify that the study aims to explore the relationship between serum zinc levels and the effectiveness of ICIs.

Response Thank you very much for your comments on our article, which makes it more prominent and bright.

3-Introduction

Line 141 : Please add "efficacy of" before "immune checkpoint inhibition". The revised sentence should read: "This study aims to explore the relationship between zinc levels and the efficacy of immune checkpoint inhibition.".

Response Thank you for your careful review. We have modified it as required.

4-Results

Line 319 : Please consider moving the paragraph about the calibration curve to the beginning of this section « This curve can evaluate the degree of calibration of the model by determining whether the probability of the model's output is consistent with the actual observed probability». It will provide context for how the predictive accuracy of the models was assessed before discussing the nomograms and results.

Response We have adjusted the order of paragraphs to make it easier for readers to follow.

5-Discussion

You have significantly improved the discussion by adding strong arguments to support the results. I only have a few minor suggestions for further refinement.

Line 332: advanced or metastatic cancer (Please add metastatic).

Line 351: Please remove "immunotherapy" since "ICIs" already refers to it.

Line 410: The phrase "In our mouse model of drug-induced lung tumors" seems to reference another author’s work, not the current study. Please, It should be clarified to avoid confusion.

Response Thanks for your correction, and your comments are very helpful to us.

Line 430: Please remove the correlation between zinc levels and PFS, as there is no significant difference between low and high zinc levels in PFS (median PFS of 9.0 months for both; p=0.7190, Fig 2B).

Response We are sorry for the inconvenience caused by our mistake. We have deleted the " progression-free survival ".

---

## [Decision Letter · Decision Letter 2]

Serum Zinc as A Biomarker to Predict the Efficacy of Immune Checkpoint Inhibitors in Cancers

PONE-D-24-44913R2

Dear Dr. Fu,

We’re pleased to inform you that your manuscript has been judged scientifically suitable for publication and will be formally accepted for publication once it meets all outstanding technical requirements.

Kind regards,

Jie Yang, M.D.

Guest Editor

PLOS ONE

Additional Editor Comments (optional):

Thanks for the authors' efforts to comprehensively improve your manuscript according to editor's and reviewers' comments. I am pleased to inform you that your paper can be accepted for publication now. Thanks for the chance to assess your interesting and important work. Additionally, many thanks for all the reviewers' precious inputs.

Reviewers' comments:

Reviewer's Responses to Questions

**Comments to the Author**

Reviewer #1: All comments have been addressed

Reviewer #4: All comments have been addressed

2. Is the manuscript technically sound, and do the data support the conclusions?

Reviewer #1: Yes

Reviewer #4: Yes

3. Has the statistical analysis been performed appropriately and rigorously?

Reviewer #1: Yes

Reviewer #4: Yes

4. Have the authors made all data underlying the findings in their manuscript fully available?

Reviewer #1: Yes

Reviewer #4: Yes

5. Is the manuscript presented in an intelligible fashion and written in standard English?

Reviewer #1: Yes

Reviewer #4: Yes

Reviewer #1: The author has satisfactorily addressed the questions I raised in the revised manuscript. I have no additional comments.

Reviewer #4: The authors have provided satisfactory responses, and the revised manuscript is now clearer and more readable. No further major revisions are required.

**Do you want your identity to be public for this peer review?** For information about this choice, including consent withdrawal, please see our Privacy Policy

Reviewer #1: No

Reviewer #4: **Yes: ** Badiaa Batlamous

---

## [Editor Report · Acceptance letter]

PONE-D-24-44913R2

PLOS ONE

Dear Dr. Fu,

I'm pleased to inform you that your manuscript has been deemed suitable for publication in PLOS ONE. Congratulations! Your manuscript is now being handed over to our production team.

Kind regards,

on behalf of

Dr. Jie Yang

Guest Editor

PLOS ONE